# The Effects of Aflatoxin B_1_ on Liver Cholestasis and Its Nutritional Regulation in Ducks

**DOI:** 10.3390/toxins16060239

**Published:** 2024-05-24

**Authors:** Aimei Yu, Huanbin Wang, Qianhui Cheng, Shahid Ali Rajput, Desheng Qi

**Affiliations:** 1Department of Animal Nutrition and Feed Science, College of Animal Science and Technology, Huazhong Agricultural University, Wuhan 430070, China; yam@webmail.hzau.edu.cn (A.Y.); whbin@webmail.hzau.edu.cn (H.W.); cqh@webmail.hzau.edu.cn (Q.C.); 2Faculty of Veterinary and Animal Science, Muhammad Nawaz Shareef University of Agriculture Multan, Multan 60000, Pakistan; shahid.ali@mnsuam.edu.pk

**Keywords:** AFB_1_, duck, cholestasis, cholestyramine, atorvastatin calcium, taurine, emodin, bile acid

## Abstract

The aim of this study was to investigate the effects of aflatoxin B_1_ (AFB_1_) on cholestasis in duck liver and its nutritional regulation. Three hundred sixty 1-day-old ducks were randomly divided into six groups and fed for 4 weeks. The control group was fed a basic diet, while the experimental group diet contained 90 μg/kg of AFB_1_. Cholestyramine, atorvastatin calcium, taurine, and emodin were added to the diets of four experimental groups. The results show that in the AFB_1_ group, the growth properties, total bile acid (TBA) serum levels and total superoxide dismutase (T-SOD), glutathione peroxidase (GSH-Px), and glutathione (GSH) liver levels decreased, while the malondialdehyde (MDA) and TBA liver levels increased (*p* < 0.05). Moreover, AFB_1_ caused cholestasis. Cholestyramine, atorvastatin calcium, taurine, and emodin could reduce the TBA serum and liver levels (*p* < 0.05), alleviating the symptoms of cholestasis. The qPCR results show that AFB_1_ upregulated *cytochrome P450 family 7 subfamily A member 1* (*CYP7A1*) and *cytochrome P450 family 8 subfamily B member 1* (*CYP8B1*) gene expression and downregulated *ATP binding cassette subfamily B member 11* (*BSEP*) gene expression in the liver, and taurine and emodin downregulated *CYP7A1* and *CYP8B1* gene expression (*p* < 0.05). In summary, AFB_1_ negatively affects health and alters the expression of genes related to liver bile acid metabolism, leading to cholestasis. Cholestyramine, atorvastatin calcium, taurine, and emodin can alleviate AFB_1_-induced cholestasis.

## 1. Introduction

Fungal toxins have caused significant harm to human health and livestock production [1]. Aflatoxin is a type of fungal toxin mainly produced by *Aspergillus flavus* [2]. It is widely present in various food crops and may cause contamination during crop maturation, harvesting, and storage [3]. There are various types of aflatoxin, among which is the B_1_ type (AFB_1_) has the most potent toxicity [4]. AFB_1_ poisoning can lead to a decrease in animal growth and negatively affect health [5,6]. The liver is the main target organ for AFB_1_ in animals, and ducklings are very sensitive to AFB_1_ [5,6]. At present, there are no reports of AFB_1_ causing liver cholestasis in ducklings.

Bile is synthesized in the liver; its main component is bile acids [7]. The liver synthesizes bile acids from cholesterol as a raw material [8]. There are two pathways for the synthesis of bile acids, namely the classic pathway mediated by cholesterol-7-α-hydroxylase (*CYP7A1*) and the alternative pathway mediated by *sterol-27-hydroxylase* (*CYP27A1*) [9]. The synthesized bile acids are discharged into the intestine through the bile duct, a process that is regulated by the bile salt efflux pump (BESP) [10]. After bile acids enter the gut, most of them are reabsorbed in the ileum and enter the bloodstream, while a small portion is excreted in the feces [9]. The reabsorbed bile acids are circulated through the bloodstream and returned to the liver for reuse [9].

Excessive synthesis of bile acids in the liver or obstruction of bile acid excretion can cause cholestasis [11]. Cholestyramine promotes the excretion of bile acids via feces, which can indirectly stimulate the efflux of bile acids from the liver [12]. Previously, it was reported that cholestyramine can reduce the concentration of bile acids and chenodeoxycholic acid in the serum of patients with cholestasis [13]. Atorvastatin calcium can inhibit liver cholesterol synthesis, reduce the raw materials available for bile acid synthesis, and indirectly inhibit bile acid synthesis [14]. Li et al. reported that atorvastatin calcium could further increase TBA levels in the ileum and feces of atherosclerosis model mice [15]. Taurine and emodin have antioxidant effects and may relieve AFB_1_ poisoning [16,17]. In addition, taurine can promote the efflux of bile acids from the liver, and emodin inhibits the production of bile acids [18,19]. Moreover, taurine deficiency can lead to a decrease in the content of bile acids in cat bile [20]. A recent study revealed that emodin can upregulate the expression of *BSEP* genes and proteins, thereby promoting bile excretion [21]. At present, there are no reports on whether these four substances can alleviate the effects of AFB_1_-induced liver cholestasis in ducklings.

This study explored the biological mechanism and nutritional regulation of AFB_1_-induced liver cholestasis by evaluating the effects of AFB_1_ on production performance, antioxidant indicators, total bile acid content in the liver, liver cholestasis, and the expression of genes related to bile acid metabolism in the liver of ducks. In addition, it investigated the ability of cholestyramine, atorvastatin calcium, taurine, and emodin to alleviate AFB_1_ poisoning and cholestasis. The findings from this study can provide a reference for the prevention and treatment of AFB_1_ poisoning in livestock and poultry.

## 2. Results

### 2.1. Production Performance and Organ Weight Changes in Ducks

The effects of the treatments on the production performance and organ weights of ducks are shown in Table 1. Compared with the control group, AFB_1_ significantly reduced the body weight (BW), average daily weight gain (ADWG), and average daily feed intake (ADFI) of ducks at 1–2 and 3–4 weeks and increased the liver and spleen weights at week 4 (*p* < 0.05). Compared with the AFB_1_ group, cholestyramine, taurine, and emodin significantly increased the BW and ADWG of ducks at 1–2 weeks (*p* < 0.05). At 3–4 weeks, compared with the AFB_1_ group, cholestyramine and taurine significantly increased the BW and ADWG of ducks, while emodin significantly increased the BW of ducks (*p* < 0.05). In addition, cholestyramine, taurine, and emodin significantly reduced the liver weight (*p* < 0.05). Meanwhile, compared with the AFB_1_ group, taurine and emodin significantly reduced the spleen weight (*p* < 0.05). Atorvastatin calcium did not significantly improve the production performance of ducks, but it did lessen the liver weight in week 4 (*p* < 0.05).

### 2.2. Changes in Liver Antioxidant Indicators

The effects of the treatments on the antioxidant indicators in duck liver are shown in Table 2. Compared with the control group, AFB_1_ significantly increased the MDA content and decreased the T-SOD, GSH-Px, and GSH contents (*p* < 0.05). Compared with the AFB_1_ group, in week 2, the MDA content in the AFB_1_ + taurine and AFB_1_ + emodin groups showed a downward trend, while the T-SOD, GSH-Px, and GSH contents showed an upward trend. In week 4, taurine significantly reduced the MDA content and increased the T-SOD, GSH-Px, and GSH contents (*p* < 0.05). Emodin significantly reduced the MDA content, increased the GSH content (*p* < 0.05), and showed an upward trend in the T-SOD and GSH-Px contents.

### 2.3. Histological Analysis of the Liver

The effects of the treatments on the morphology and steatosis of duck liver tissue are shown in Figure 1. Hematoxylin and eosin (H&E) staining showed that the liver morphology of the control group was normal. In contrast, the liver cells of the AFB_1_ group showed swelling and an extremely uneven distribution of nuclei. Compared with the AFB_1_ group, the liver cell swelling in the AFB_1_ + taurine and AFB_1_ + emodin groups was evidently decreased. Additionally, oil red O staining showed no abnormalities in the liver of the control group, while the liver of the AFB_1_ group showed significant steatosis, denoted by large red areas of lipid staining. Strikingly, taurine and emodin treatment alleviated steatosis in the liver of ducks induced by AFB_1_, with a reduction in the red area in the field of view. Compared with the AFB_1_ group, there was no significant improvement in the liver of the AFB_1_ + cholestyramine and AFB_1_ + atorvastatin calcium groups.

### 2.4. Changes in Biochemical Indicators in Duck Serum

The effects of the treatments on the biochemical indicators in duck serum are shown in Table 3. Compared with the control group, AFB_1_ significantly increased the aspartate transaminase (AST), alanine transaminase (ALT), TBA, and total bilirubin (TBiL) levels and significantly reduced the total protein (TP) and albumin (ALB) levels (*p* < 0.05). In week 2, compared with the AFB_1_ group, taurine and emodin treatments significantly reduced the AST and TBA levels (*p* < 0.05), and the TBiL levels also showed a downward trend. In addition, cholestyramine significantly reduced the TBA levels (*p* < 0.05). In week 4, compared with the AFB_1_ group, cholestyramine and atorvastatin calcium significantly reduced the TBA and TBiL levels (*p* < 0.05). At the same time, taurine significantly increased the TP and ALB levels (*p* < 0.05), while the AST, ALT, TBA, and TBiL levels showed a descending trend. Emodin significantly reduced the TBA levels (*p* < 0.05), while the AST, ALT, and TBiL levels showed a decreasing trend.

### 2.5. The Degree of Liver Cholestasis in the Treatment Groups

The effect of the treatments on the cholestasis of duck liver in week 4 is shown in Figure 2. The control group presented a normal liver color, indicating no cholestasis. However, the liver of the AFB_1_ group had a greenish color, indicative of cholestasis. The addition of cholestyramine, atorvastatin calcium, taurine, and emodin to the diet of the experimental group alleviated the liver cholestasis of duck to varying degrees induced by AFB_1_.

### 2.6. Changes in the Total Bile Acid Content in the Liver

The effect of the treatments on the total bile acid content in the liver of the ducks is shown in Figure 3. Compared with the control group, AFB_1_ significantly increased the TBA content in weeks 2 and 4 (*p* < 0.05). Compared with the AFB_1_ group, cholestyramine, atorvastatin calcium, taurine, and emodin treatments significantly reduced the TBA content in the liver of ducks (2 and 4 weeks). 

### 2.7. Changes in the Expression Levels of Liver-Bile-Acid-Metabolism-Related Genes

The effects of the treatments on the expression levels of genes related to liver bile acid metabolism are shown in Figure 4. AFB_1_ significantly upregulated the expression of *CYP7A1* and *CYP8B1* in the liver and downregulated the expression of *BSEP*, *nuclear receptor subfamily 0 group B member 2* (*SHP*), and *solute carrier organic anion transporter family member 1A2* (*OATP*) (*p* < 0.05). Compared with the AFB_1_ group, the expression of *CYP7A1* and *CYP8B1* in the AFB_1_ + taurine and AFB_1_ + emodin groups was significantly downregulated (*p* < 0.05), while the expression of *CYP7A1* and *CYP8B1* in the AFB_1_ + cholestyramine and AFB_1_ + atorvastatin calcium groups showed a downward trend. Compared with the AFB_1_ group, the AFB_1_ + atorvastatin calcium group showed significant upregulation of *BSEP* and *OATP* (*p* < 0.05), while the AFB_1_ + emodin group showed a trend for increased *BSEP* expression. No significant gene expression changed in the AFB_1_ + cholestyramine group. 

## 3. Discussion

Numerous studies have shown that AFB1 can cause growth retardation and liver damage in animals, with symptoms of poisoning mainly manifested as liver toxicity and oxidative stress [5,22,23]. Gao et al. [22] reported that AFB1 increased the levels of transaminases in chicken serum, decreased the antioxidant capacity of chickens, and caused swelling and necrosis of chicken liver cells. Wang et al. [5] reported that AFB1 not only leads to a decrease in growth performance and antioxidant levels in ducklings but also causes cellular swelling, steatosis, and nuclear shrinkage in the liver of ducklings. In addition, Liu et al. [23] reported that AFB1 reduced body weight and T-AOC and GSH-Px liver levels in mice. The present study showed that AFB1 can lead to a decrease in duck production performance and liver antioxidant levels, as well as symptoms such as liver cell swelling and steatosis, which is consistent with previous research results. AFB1 in feed poses a threat to the health of poultry and decreases the quality of their meat [24].

Bile acids are synthesized by the liver and secreted into the intestine to digest and absorb nutrients [7,8,9]. Bile acids also participate in the regulation of the structure of gut microbiota; thus, they are closely related to animal health [25]. There are currently no reports of AFB1 causing cholestasis in the duck liver. AFB_1_ can upregulate *CYP7A1* gene expression in the mouse liver, promoting bile acid synthesis and thereby increasing bile acid content in the mouse serum, liver, and feces [26]. Furthermore, AFB_1_ can cause the obstruction of bile secretion in rats [27]. In addition, rats orally administered 25 μg/kg of AFB_1_ showed an increase in bile acids in feces, while oral administration of 5 μg/kg AFB_1_ reduced the content of deoxycholic acid, a type of bile acid, in feces [28]. This result suggests that different doses of AFB_1_ may have different effects on the metabolism of bile acids. The present study demonstrated that AFB_1_ can cause liver cholestasis and increase the TBA serum and liver levels. *CYP7A1* and *CYP8B1* gene expression in the liver was upregulated, a finding similar to these reported results. Therefore, it is speculated that the AFB_1_-mediated increase in liver bile acid synthesis leads to liver cholestasis.

*BSEP* is a key gene that controls the excretion of bile acids in liver cells [29,30,31]. This gene’s absence or abnormal expression can lead to cholestasis [29,30,31]. Currently, there are no relevant reports on the regulation of the *BSEP* gene by AFB_1_. The present study showed that AFB_1_ downregulates *BSEP* gene expression in the liver. This downregulation can lead to obstruction of bile acid efflux in liver cells [29], and the significant increase in the TBA liver content in this study also supports this view. Therefore, abnormal *BSEP* gene expression may also be related to AFB_1_-induced liver cholestasis.

Cholestyramine, as a bile acid chelator, can bind to bile acids in the intestine, promoting the excretion of bile acids via the feces and reducing the reabsorption of bile acids in the ileum [12]. In clinical practice, cholestyramine is mainly used to treat hypercholesterolemia and cholestasis [13,32]. The present study showed that cholestyramine can alleviate the symptoms of cholestasis caused by AFB_1_ and reduce the bile acid content in duck liver and serum, which is similar to the reported results. In addition, this study revealed that cholestyramine improves the decline in duck growth performance caused by AFB_1_, which may be related to its ability to alleviate cholestasis caused by AFB_1_ and reduce the harmful effects of cholestasis. The specific reasons still need further research.

The liver synthesizes bile acids, and cholesterol is the raw material for bile acid synthesis [8]. Atorvastatin calcium is a lipid-lowering drug that can inhibit cholesterol synthesis in the liver [14]. Lastuvkova et al. reported that atorvastatin calcium increased the deoxycholic acid content in the plasma of nonalcoholic steatohepatitis model mice, promoting bile secretion and the excretion of bile acids in feces [33]. The present study demonstrated that atorvastatin calcium alleviates cholestasis and reduces the content of bile acids in the liver but does not change the TBA serum levels. The results of this study differ from previous reports, a phenomenon that may be related to the combined effect of atorvastatin calcium and AFB_1_; the specific reasons need further investigation. In addition, atorvastatin calcium does not alleviate the decline in duck growth performance caused by AFB_1_, which may be due to the involvement of cholesterol in duck growth and development [34]. The decrease in cholesterol synthesis affects the normal growth of ducklings.

Bile acids have two forms, free bile acids and conjugated bile acids, and taurine can combine with free bile acids in the liver to form taurine-conjugated bile acids, promoting the excretion of bile acids [9,18]. Taurine can promote the excretion of bile acids in rats and reduce cholesterol in serum [35]. The current study showed that taurine can reduce the total bile acid content in serum and liver and alleviate cholestasis caused by AFB_1_, similar to the previous reports. In addition, this study revealed that taurine can enhance the antioxidant capacity of ducks and alleviate AFB_1_-induced growth inhibition and liver damage in ducks. A previous study reported that taurine can alleviate AFB_1_-induced liver injury in rats by enhancing antioxidant capacity and inhibiting mitochondrial-mediated cell apoptosis [36]; our results are similar to this report.

Emodin has antioxidant and anti-inflammatory effects and can promote bile acid excretion [17,19]. Emodin can alleviate α-naphthyl isothiocyanate-induced cholestasis in mice [37]. Similarly, in the present study, emodin can reduce the total bile acid content in serum and liver, alleviate cholestasis caused by AFB_1_, and upregulate *BSEP* gene expression in the liver. There are currently no reports on the effect of emodin on AFB_1_. Xia et al. reported that emodin can alleviate acute pancreatitis through its antioxidant and anti-inflammatory effects [38]. This study also demonstrated that emodin enhances the antioxidant capacity of ducks, which is consistent with our reported results. After the increase in antioxidant capacity, the poisoning symptoms caused by AFB_1_ were alleviated, so the growth performance of ducks was also improved compared with the challenged group. AFB_1_-induced liver cholestasis may be related to abnormal synthesis and excretion of bile acids. Reducing bile acid synthesis and promoting bile acid efflux are effective ways to alleviate cholestasis.

## 4. Conclusions

In summary, AFB_1_ disrupts the normal growth of ducks, negatively affects the liver, and leads to cholestasis. The abnormal expression of bile acid metabolism-related genes in the liver may be an important reason for AFB_1_-induced cholestasis in duck liver. Cholestyramine can promote the efflux of bile acids, while atorvastatin calcium, taurine, and emodin can reduce the synthesis of bile acids. These four compounds can alleviate AFB_1_-mediated liver cholestasis. This study can provide a reference for the prevention and treatment of AFB_1_ poisoning in livestock and poultry.

## 5. Materials and Methods

### 5.1. Duck, Treatments, and Sample Collection

This experiment was approved by the Animal Ethics and Use Committee of Huazhong Agricultural University (approval number HZAUDU-2024-0005). Three-hundred sixty 1-day-old cherry valley meat ducks were randomly divided into 6 groups: control, AFB_1_, AFB_1_ + cholestyramine, AFB_1_ + atorvastatin calcium, AFB_1_ + taurine, and AFB_1_ + emodin. Each group had 6 replicates, with 10 ducks per replicate. All ducks are kept in stainless steel cages and can freely obtain a basic diet (Appendix A) and water. Before the formal experiment began, the duck house was fumigated and disinfected with potassium permanganate and formalin. AFB_1_ is derived from the fermentation products of *Aspergillus flavus* cultured in the laboratory. Cholestyramine, atorvastatin calcium, taurine, and emodin were purchased from Shanghai Macklin Biochemical Co., Ltd. (Shanghai, China; purity ≥ 98%). The doses added to their diet were AFB_1_, 90 μg/kg; cholestyramine, 6000 mg/kg; atorvastatin calcium, 75 mg/kg; taurine, 1500 mg/kg; and emodin, 400 mg/kg. After adding AFB1 to the basic diet, the actual content of AFB1 was determined using a commercial reagent kit (Romer Labs, Tulln an der Donau, Austria). The actual content of AFB1 in a uniformly mixed diet is 87 μg/kg. The ducks were raised for a total of 4 weeks, and their production performance was recorded during the feeding period. Blood samples were collected from the subwing vein at weeks 2 and 4, and serum was separated using the method reported by Wang et al. [13]. The serum was stored in a −80 °C freezer for further analysis. At the end of the experiment, the ducks were euthanized using the acute blood loss method, and the liver and spleen weights were recorded before taking photos of the liver.

### 5.2. Determination of Production Performance

At the beginning of the experiment, the average weight of each group of ducks was determined and recorded as the initial weight. Then, the average weight of each group of ducks was measured at weeks 2 and 4. The average daily weight gain was calculated based on the initial weight, the average weight at week 2, the average weight at week 4, and the number of days of feeding. The weight of feed fed each week and the remaining weight of feed were recorded to calculate the average daily feed intake. The ratio of the average daily feed intake to the average daily weight gain was recorded as the feed conversion rate.

### 5.3. Determination of Liver Antioxidant Indicators

The liver was lysed using RIPA lysis buffer (Beijing Solarbio Science & Technology Co., Ltd., Beijing, China) to obtain liver tissue homogenate. The protein concentration was measured by using the BCA Protein Assay Kit (Beyotime Biotechnology Co., Ltd., Wuhan, China). The T-SOD, MDA, GSH Px, GSH, and TBA contents in the homogenate were measured by using a commercial kit (Nanjing Jiancheng Biotechnology Co., Ltd., Nanjing, China), following the manufacturer’s instructions.

### 5.4. H&E Staining

After fixing liver tissue blocks with paraformaldehyde and embedding them in paraffin, sections were cut and stained with H&E.

### 5.5. Oil Red O Staining

The liver tissue blocks fixed with paraformaldehyde were dehydrated with sucrose solution and then frozen. Sections were cut and stained with oil red O and hematoxylin.

### 5.6. Determination of Serum Biochemical Indicators

The AST, ALT, TP, ALB, TBA, and TBiL serum levels were measured using a fully automated biochemical analyzer (Beckman, CA, USA).

### 5.7. Determination of the TBA Content in the Liver

A commercially available reagent kit (Nanjing Jiancheng Biotechnology Co., Ltd.) was used to determine the TBA content in the liver, following the manufacturer’s guidelines.

### 5.8. RNA Extraction, cDNA Synthesis, and Real-Time qPCR

Trizol (ABclonal Technology Co., Ltd., Wuhan, China) was used to extract RNA from the liver. ABScript III RT Master Mix for qPCR with gDNA Remover and 2X Universal SYBR Green Fast qPCR Mix (ABclonal Technology Co., Ltd.) was used to synthesize cDNA and perform real-time qPCR, following the manufacturer’s instructions. The primers for real-time qPCR are listed in Appendix A. All genes’ Cq values were normalized using the GAPDH Cq value as a reference, and the results were analyzed using the 2^−ΔΔCt^ method.

### 5.9. Statistical Analysis

The experimental results were analyzed using SPSS 27.0 (IBM Corp., Armonk, NY, USA) and are presented as the mean ± standard deviation. One-way analysis of variance and the t-test were to evaluate intergroup differences, where *p* < 0.05 indicated a significant difference.

## Figures and Tables

**Figure 1 toxins-16-00239-f001:**
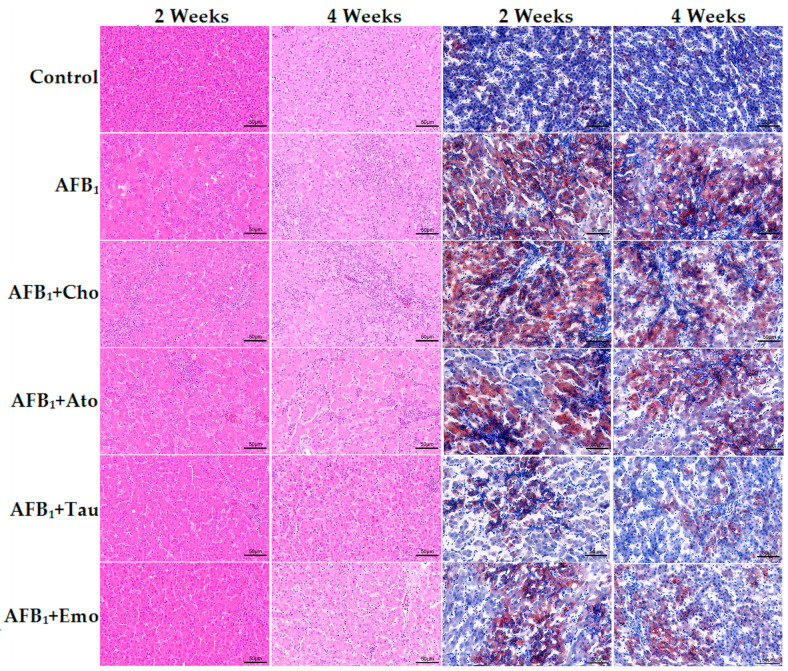
Hematoxylin and eosin (H&E) and oil red O staining of liver tissue (200× magnification). “AFB_1_ + Cho” = AFB_1_ + cholestyramine; “AFB_1_ + Ato” = AFB_1_ + atorvastatin calcium; “AFB_1_ + Tau” = AFB_1_ + taurine; “AFB_1_ + Emo” = AFB_1_ + emodin.

**Figure 2 toxins-16-00239-f002:**
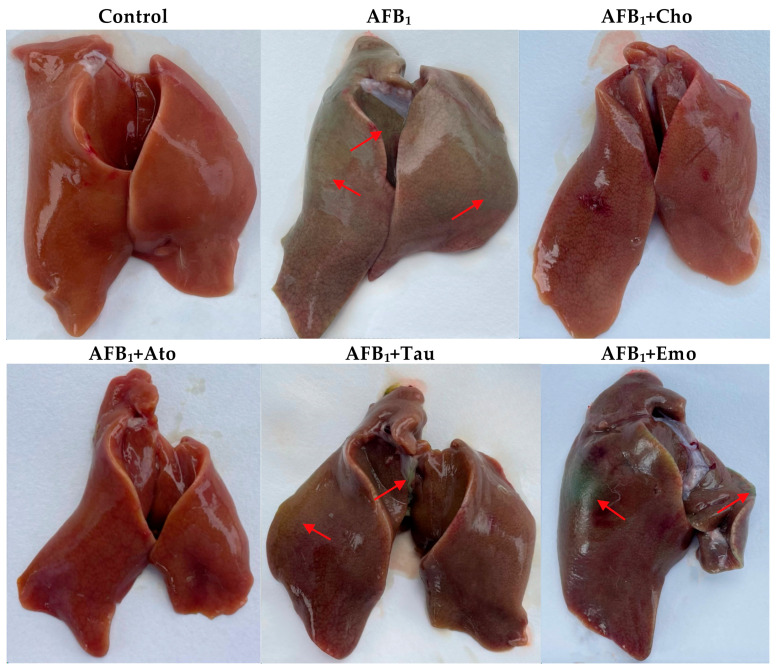
Liver anatomy map. Red arrow: location of cholestasis. “AFB_1_ + Cho” = AFB_1_ + cholestyramine; “AFB_1_ + Ato” = AFB_1_ + atorvastatin calcium; “AFB_1_ + Tau” = AFB_1_ + taurine; “AFB_1_ + Emo” = AFB_1_ + emodin.

**Figure 3 toxins-16-00239-f003:**
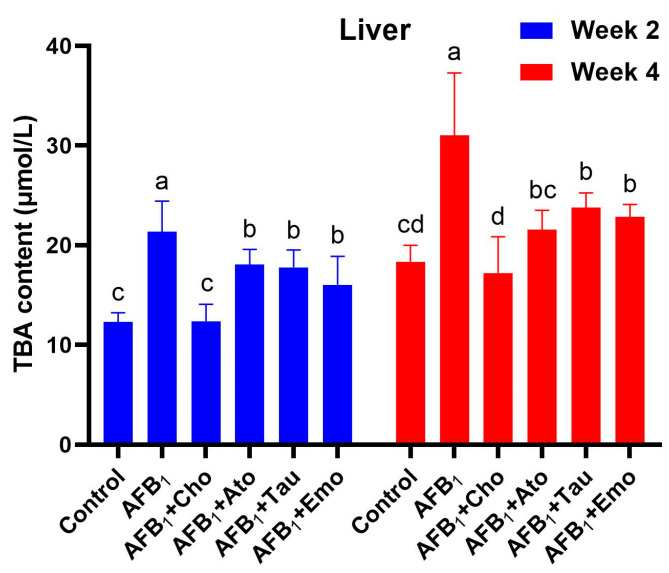
The total bile acid (TBA) content in the liver (*n* = 6). ^a–d^ Different lowercase letters indicate significant differences between the groups (*p* < 0.05). “AFB_1_ + Cho” = AFB_1_ + cholestyramine; “AFB_1_ + Ato” = AFB_1_ + atorvastatin calcium; “AFB_1_ + Tau” = AFB_1_ + taurine; “AFB_1_ + Emo” = AFB_1_ + emodin.

**Figure 4 toxins-16-00239-f004:**
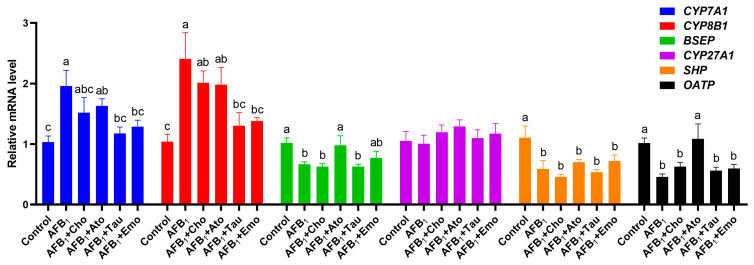
Changes in the expression of genes related to bile acid metabolism (n = 6). ^a–c^ Different lowercase letters indicate significant differences between the groups (*p* < 0.05). “AFB_1_ + Cho” = AFB_1_ + cholestyramine; “AFB_1_ + Ato” = AFB_1_ + atorvastatin calcium; “AFB_1_ + Tau” = AFB_1_ + taurine; “AFB_1_ + Emo” = AFB_1_ + emodin.

**Table 1 toxins-16-00239-t001:** Production performance and weight of the liver and spleen ^1^.

Items ^2^	Control	AFB_1_	AFB_1_ + Cho	AFB_1_ + Ato	AFB_1_ + Tau	AFB_1_ + Emo
IW (g)	54.26 ± 0.69	54.27 ± 0.60	54.18 ± 0.75	54.15 ± 0.57	54.24 ± 0.67	54.14 ± 0.60
1–2 weeks						
BW (g)	438.86 ± 17.74 ^a^	351.30 ± 8.70 ^c^	392.52 ± 17.87 ^b^	363.97 ± 8.86 ^c^	404.31 ± 23.59 ^b^	388.92 ± 10.86 ^b^
ADWG (g)	27.47 ± 1.31 ^a^	21.22 ± 0.63 ^c^	24.17 ± 1.28 ^b^	22.13 ± 0.62 ^c^	25.01 ± 1.66 ^b^	23.91 ± 0.78 ^b^
ADFI (g)	50.71 ± 3.63 ^a^	42.50 ± 1.43 ^b^	46.67 ± 5.13 ^ab^	41.53 ± 5.42 ^b^	46.81 ± 5.71 ^ab^	46.42 ± 5.25 ^ab^
FCR	1.85 ± 0.12	2.00 ± 0.07	1.93 ± 0.18	1.88 ± 0.24	1.87 ± 0.17	1.94 ± 0.21
3–4 weeks						
BW(g)	1365.58 ± 56.40 ^a^	1082.15 ± 76.64 ^d^	1220.16 ± 65.88 ^b^	1113.13 ± 88.91 ^cd^	1249.74 ± 89.56 ^b^	1191.24 ± 65.51 ^bc^
ADWG (g)	66.19 ± 4.59 ^a^	52.20 ± 5.36 ^d^	59.12 ± 5.18 ^bc^	53.51 ± 6.54 ^cd^	60.39 ± 5.33 ^ab^	57.31 ± 4.64 ^bcd^
ADFI (g)	145.47 ± 9.15 ^a^	121.16 ± 21.29 ^bc^	127.98 ± 8.87 ^b^	112.00 ± 7.37 ^c^	131.15 ± 11.21 ^ab^	132.38 ± 11.80 ^ab^
FCR	2.20 ± 0.06	2.33 ± 0.40	2.17 ± 0.06	2.12 ± 0.29	2.17 ± 0.10	2.18 ± 0.27
Week 4						
Liver (g)	32.42 ± 4.79 ^b^	43.53 ± 10.18 ^a^	34.46 ± 5.46 ^b^	31.41 ± 3.73 ^b^	32.08 ± 4.45 ^b^	27.93 ± 7.31 ^b^
Spleen (g)	1.32 ± 0.35 ^b^	2.09 ± 0.84 ^a^	1.66 ± 0.26 ^ab^	2.09 ± 0.78 ^a^	1.36 ± 0.25 ^b^	1.29 ± 0.39 ^b^

^1^ The values are expressed as the mean ± standard deviation (n = 6). Means with different superscripts differ significantly (*p* < 0.05). “AFB_1_ + Cho” = AFB_1_ + cholestyramine; “AFB_1_ + Ato” = AFB_1_ + atorvastatin calcium; “AFB_1_ + Tau” = AFB_1_ + taurine; “AFB_1_ + Emo” = AFB_1_ + emodin. ^2^ “IW” = initial weight; “BW” = body weight; “ADWG” = average daily weight gain; “ADFI” = average daily feed intake; “FCR” = feed conversion rate.

**Table 2 toxins-16-00239-t002:** Antioxidant indicators in the liver ^1^.

Items	Control	AFB_1_	AFB_1_ + Cho	AFB_1_ + Ato	AFB_1_ + Tau	AFB_1_ + Emo
Week 2						
T-SOD (U/mg protein)	100.41 ± 16.65 ^a^	77.94 ± 5.65 ^b^	74.36 ± 6.28 ^b^	53.85 ± 7.39 ^c^	84.78 ± 11.26 ^b^	88.88 ± 19.70 ^ab^
MDA (nmol/ mg protein)	1.90 ± 0.22 ^b^	2.77 ± 0.60 ^a^	2.41 ± 0.48 ^ab^	2.41 ± 0.35 ^ab^	2.26 ± 0.47 ^ab^	1.94 ± 0.38 ^b^
GSH-Px (U/mg protein)	57.34 ± 7.14 ^a^	42.51 ± 7.65 ^b^	46.00 ± 5.77 ^b^	42.17 ± 6.08 ^b^	50.69 ± 7.13 ^ab^	51.08 ± 9.18 ^ab^
GSH (U/mg protein)	37.33 ± 2.85 ^a^	24.84 ± 4.09 ^c^	30.12 ± 4.47 ^bc^	33.06 ± 9.83 ^ab^	26.95 ± 3.29 ^bc^	28.13 ± 4.00 ^bc^
Week 4						
T-SOD (U/mg protein)	119.56 ± 6.06 ^a^	90.70 ± 12.30 ^b^	102.51 ± 13.70 ^ab^	105.33 ± 17.25 ^ab^	119.47 ± 12.21 ^a^	98.88 ± 16.05 ^b^
MDA (nmol/ mg protein)	1.70 ± 0.10 ^b^	2.46 ± 0.48 ^a^	1.96 ± 0.47 ^b^	1.73 ± 0.46 ^b^	1.70 ± 0.21 ^b^	1.73 ± 0.58 ^b^
GSH-Px (U/mg protein)	69.42 ± 10.79 ^a^	44.80 ± 5.52 ^c^	51.30 ± 17.29 ^bc^	51.06 ± 5.60 ^bc^	63.10 ± 13.37 ^ab^	57.34 ± 7.63 ^abc^
GSH (U/mg protein)	39.46 ± 2.03 ^a^	29.68 ± 5.19 ^b^	34.76 ± 4.28 ^ab^	36.52 ± 6.21 ^a^	36.97 ± 1.98 ^a^	39.87 ± 5.96 ^a^

^1^ The values are expressed as the mean ± standard deviation (n = 6). Means with different superscripts differ significantly (*p* < 0.05). “AFB_1_ + Cho” = AFB_1_ + cholestyramine; “AFB_1_ + Ato” = AFB_1_ + atorvastatin calcium; “AFB_1_ + Tau” = AFB_1_ + taurine; “AFB_1_ + Emo” = AFB_1_ + emodin.

**Table 3 toxins-16-00239-t003:** Biochemical indicators in the serum ^1^.

Items	Control	AFB_1_	AFB_1_ + Cho	AFB_1_ + Ato	AFB_1_ + Tau	AFB_1_ + Emo
Week 2						
AST (U/L)	33.25 ± 3.41 ^d^	59.17 ± 7.01 ^a^	53.58 ± 11.11 ^ab^	46.22 ± 12.31 ^bc^	43.37 ± 7.59 ^c^	32.95 ± 2.52 ^d^
ALT (U/L)	46.05 ± 8.83	53.77 ± 5.34	46.77 ± 9.49	49.95 ± 4.85	52.18 ± 9.88	43.82 ± 6.41
TP (g/L)	31.32 ± 2.80 ^a^	20.37 ± 1.58 ^c^	23.77 ± 4.86 ^bc^	21.38 ± 1.73 ^c^	23.18 ± 2.78 ^bc^	25.13 ± 1.91 ^b^
ALB (g/L)	9.88 ± 1.30 ^a^	5.77 ± 1.05 ^c^	6.27 ± 0.86 ^bc^	6.13 ± 0.98 ^bc^	7.35 ± 0.94 ^b^	6.93 ± 1.17 ^bc^
TBA (μmol/L)	12.65 ± 1.27 ^c^	20.96 ± 3.64 ^a^	12.66 ± 1.96 ^c^	18.64 ± 1.22 ^ab^	17.77 ± 1.75 ^b^	16.78 ± 2.95 ^b^
TbiL (μmol/L)	5.83 ± 1.15 ^b^	7.43 ± 1.09 ^a^	6.29 ± 1.01 ^ab^	6.65 ± 0.87 ^ab^	6.98 ± 1.58 ^ab^	6.75 ± 1.27 ^ab^
Week 4						
AST (U/L)	33.12 ± 7.31 ^b^	43.78 ± 8.73 ^a^	38.48 ± 12.36 ^ab^	38.90 ± 3.61 ^ab^	37.85 ± 6.81 ^ab^	39.07 ± 4.51 ^ab^
ALT (U/L)	35.38 ± 7.74 ^b^	45.58 ± 8.58 ^a^	44.23 ± 6.16 ^ab^	43.35 ± 6.59 ^ab^	39.45 ± 4.56 ^ab^	37.10 ± 7.19 ^ab^
TP (g/L)	28.62 ± 1.58 ^a^	22.98 ± 2.81 ^c^	22.82 ± 1.61 ^c^	25.40 ± 2.25 ^bc^	27.67 ± 2.67 ^ab^	24.83 ± 2.1 ^bc^
ALB (g/L)	12.70 ± 0.89 ^a^	9.70 ± 0.98 ^b^	9.65 ± 0.94 ^b^	9.82 ± 1.07 ^b^	11.90 ± 1.68 ^a^	11.10 ± 2.90 ^ab^
TBA (μmol/L)	18.58 ± 2.04 ^c^	29.03 ± 6.78 ^a^	18.22 ± 5.81 ^c^	21.63 ± 1.99 ^bc^	24.92 ± 1.60 ^ab^	22.83 ± 1.27 ^bc^
TbiL (μmol/L)	5.88 ± 1.43 ^bc^	9.99 ± 0.90 ^a^	4.64 ± 1.43 ^c^	6.78 ± 1.61 ^b^	8.68 ± 0.99 ^a^	8.59 ± 2.01 ^a^

^1^ The values are expressed as the mean ± standard deviation (n = 6). Means with different superscripts differ significantly (*p* < 0.05). “AFB_1_ + Cho” = AFB_1_ + cholestyramine; “AFB_1_ + Ato” = AFB_1_ + atorvastatin calcium; “AFB_1_ + Tau” = AFB_1_ + taurine; “AFB_1_ + Emo” = AFB_1_ + emodin.

## Data Availability

The data presented in this study are available on request from the corresponding author.

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
