# Peer review of "The Effects of Aflatoxin B1 on Liver Cholestasis and Its Nutritional Regulation in Ducks"

_toxins, 2024, doi:10.3390/toxins16060239_

Round 1

Reviewer 1 Report

Comments and Suggestions for Authors

A legend with the description of abbreviations would be necessary, both in terms of malondialdehyde, serum total bile acid and other markers of oxidative stress, as well as liver damage, the same for antioxidant enzymes, and for the various gene expressions, there is no description of abbreviation in text or in separate list. Moreover a brief description of MDA and TBA would be necessary in the Introduction chapter.

In the Materials and Methods section, there is no information regarding the presence of the value of 90 µg/kg AFB1 in the feed, whether this value has been verified, recovery rate and by what method was used to determine the mycotoxin.

Comments on the Quality of English Language

Minor editing of English language required

Reviewer 2 Report

Comments and Suggestions for Authors

This is an interesting manuscript describing the modulation of aflatoxin B1 (AFB1) toxicity by some nutritional components,  namely cholestyramine, atorvastatin calcium, taurine, and emodin, on ducklings. The experimentals sound scientifically, and the data presented are acceptable - after clarifying or addressing the following revision comments:

-Introduction: P1, L2-3: There is no "parasitic Aspergillus spp". L5 and throughout the text: Use subscript "1" when referreing to "aflatoxin B1" (use B1) or "AFB1" (use AFB1). Also, is there any citation for the sentence "Previous research in our laboratory has demonstrated... liver cholestasis in ducklings"? Background information about the nutritional compounds evaluated is missing in this section. Finally, on P2, L5-6, it is indicated that taurine and emodin have antioxidant effects and may relieve AFB1 poisoning; so, what´s the novelty of the paper regarding these compounds?

-Results: P.3, section 2.2 (and elsewhere in the text): Put the definition of all abbreviations before mentioning them, like "T-SOD", "GSH", "MDA", etc.

-Discussion: P8, last sentence of the first paragraph: as no residues were determined in the experiment, what is the relevance of this subject for the discussion?

-Material & Methods: P.10, first paragraph: What is the rationale for using those levels of AFB1 and nutritional compounds? Also, did you check the level of AFB1 in the diets? This is essential for validation of the data presented in the manuscript.

Reviewer 3 Report

Comments and Suggestions for Authors

In this paper, live duck feeding experiments were conducted between the AFB1 group and the Cholestyramine, atorvastatin calcium, taurine, and emodin groups. To investigate the effects of aflatoxin B1 (AFB1) on cholestasis in the liver of ducks and its nutritional regulation, the results showed that AFB1 not only affects the normal growth of ducks, but also mediates liver cholestasis, and the other four compounds can reduce AFB1-mediated liver cholestasis. The relevant conclusions of this study can provide reference for the prevention and treatment of AFB1 poisoning in livestock and poultry. However, there are still the following problems in the article, please check and correct:

1.     It is mentioned in the preface that the previous study of the laboratory found that AFB1 in the feed can cause liver cholestasis in ducklings. If there are published literatures, they should be cited; otherwise, relevant experimental data should be supplemented.

2.     There are problems with the structure of the article. It is suggested to place "5. Materials and Methods" after "1. Introduction" to make the article more smooth and clear;

3.     The results of some forms are too messy, it is suggested that the contents of the form should be classified by lines, or divided into different forms, so that the data comparison is more obvious; (Table 1-3)

4.     Please pay attention to the format of the article. For example, "Figure 1." should be abbreviated as "Fig.1"; And the notes of each abbreviation symbol are messy, some notes are not obvious enough, please optimize;

5.     The discussion part cites a large number of research examples to explain the compounds in the experiment, which makes the content of this part lengthy and redundant. It is suggested that some examples should be added to the preface;

6.     Are the living Spaces of all experimental live ducks mentioned in 5.1 sterilized and sterilized? If not, how to ensure that the experimental subjects will not be affected by external factors?

7.     In this paper, the measurement period of each experimental object was set as "Week 2 and Week 4". In order to ensure the accuracy of experimental data, could the test period be further refined, such as "Day 3, day 7, day 14, day 21 and day 30"?

8.     Some of the literature cited in this paper is too old and lacks effectiveness. It is suggested to replace and supplement the literature of recent five years; And there are some grammar problems in the article, please check carefully and correct.

Comments on the Quality of English Language

Moderate editing of English language required

Round 2

Reviewer 1 Report

Comments and Suggestions for Authors

I have no further comments to add, I generally agree with the changes made.

Comments on the Quality of English Language

Minor editing of English language required

Reviewer 3 Report

Comments and Suggestions for Authors

accept

Comments on the Quality of English Language

accept